# The Reduction Factor of Pultrude Glass Fibre-Reinforced Polyester Composite Cross-Arm: A Comparative Study on Mathematical Modelling for Life-Span Prediction

**DOI:** 10.3390/ma16155328

**Published:** 2023-07-29

**Authors:** Mohd Supian Abu Bakar, Agusril Syamsir, Abdulrahman Alhayek, Muhammad Rizal Muhammad Asyraf, Zarina Itam, Shaikh Muhammad Mubin Shaik, Nurhanani Abd Aziz, Tarique Jamal, Siti Aminah Mohd Mansor

**Affiliations:** 1Institute of Energy Infrastructure (IEI), College of Engineering, Universiti Tenaga Nasional, Kajang 43000, Malaysia; mohd.supian@uniten.edu.my (M.S.A.B.); izarina@uniten.edu.my (Z.I.); tarique.jamal@uniten.edu.my (T.J.); aminah@uniten.edu.my (S.A.M.M.); 2Civil Engineering Department, College of Engineering, Universiti Tenaga Nasional, Kajang 43000, Malaysia; abdul.rahman@uniten.edu.my (A.A.); shaikh.muhammad@uniten.edu.my (S.M.M.S.); nurhanani.aziz@uniten.edu.my (N.A.A.); 3Centre for Advanced Composite Materials (CACM), Universiti Teknologi Malaysia, Johor Bahru 81310, Malaysia; muhammadasyraf.mr@utm.my

**Keywords:** reduction factor, pultrude glass fibre-reinforced polyester composite, mathematical model, life-span prediction, energy

## Abstract

This paper presents an experimental and numerical investigation of pultruded composite glass fibre-reinforced polymer (pGFRP) cross-arms subjected to flexural creep behaviour to assess their performance and sustainability in composite cross-arm structure applications. The primary objective of this study was to investigate the failure creep behaviour of pGFRP cross-arms with different stacking sequences. Specifically, the study aimed to understand the variations in strain rate exhibited during different stages of the creep process. Therefore, this study emphasizes a simplified approach within the experiment, numerical analysis, and mathematical modelling of three different pGFRP composites to estimate the stiffness reduction factors that determine the prediction of failure. The findings show that Findley’s power law and the Burger model projected very different strains and diverged noticeably outside the testing period. Findley’s model estimated a minimal increase in total strain over 50 years, while the Burger model anticipated PS-1 and PS-2 composites would fail within about 11 and 33 years, respectively. The Burger model’s forecasts might be more reasonable due to the harsh environment the cross-arms are expected to withstand. The endurance and long-term performance of composite materials used in overhead power transmission lines may be predicted mathematically, and this insight into material property factors can help with design and maintenance.

## 1. Introduction

Fibre-reinforced polymeric (FRP) composites, particularly in the form of transmission and distribution line cross-arms, are widely used in lattice towers. These cross-arms, made of glass fibre-reinforced polyester (GFRP) composites, experience significant creep deformation over time due to the continuous stress exerted by power cables [1,2]. Various factors contribute to this phenomenon, including shear yielding, polymer chain slippage, fibre breakage, void development, and growth [3,4,5]. To ensure the high-performance of GFRP cross-arms in lattice towers, it is crucial to conduct research on their mechanical behaviour, including creep, through empirical models, numerical analyses, or experiments under different service conditions [6,7,8,9].

The long-term performance and life-span of GFRP composite cross-arms used in power transmission systems are critical for the reliability and safety of infrastructure [6,10,11,12]. However, accurately estimating the reduction factor, which indicates structural deterioration and load-carrying capacity over time, remains a challenge [13,14,15,16,17]. Established mathematical models such as Burger, Findley, Arrhenius, and Norton–Bailey are inadequate for accurately predicting the reduction factor of GFRP cross-arms [18,19]. Therefore, a comparative study of mathematical modelling methodologies is needed to establish the best model for forecasting the reduction factor and improving life-span forecasts. This study aims to bridge this knowledge gap and enhance our understanding of the endurance and structural behaviour of composite cross-arms, enabling better maintenance and replacement decisions for power transmission systems [20,21].

Severe creep deformation caused by extreme climate conditions and biological attacks significantly reduces the longevity of composite cross-arms. The creep process typically consists of three stages: primary (quick rate), secondary (steady state), and tertiary (rapid rate to rupture) [3,4,5,22,23]. Improper creep analysis can lead to material failure without warning, potentially resulting in the catastrophic collapse of building structures [8,22,23,24]. Therefore, analysing the safe limit parameter of the cross-arm structure and establishing uniform safety criteria are essential [25,26,27,28]. This can be achieved through experimentation, empirical analysis, and computational modelling, including computer models and coupon-scale or full-scale structural tests (Table 1).

To extend the service life of composite cross-arm constructions, several strategies and improvements have been proposed, including changes in lamination order, structural enhancements, and hybrid composite structures [37,38]. For instance, Zaghloul et al. [39] discovered that surface reinforced arranged composites have 61 times the life of bulk reinforced arranged composites after being exposed to bending fatigue at 56 MPa bending stress. The integration of biaxially and orientated polymeric fibres and fillers within the matrix has shown promise in improving the mechanical characteristics of these structures [40,41,42,43]. Additionally, the use of additives in GFRP composites and the incorporation of a core structure, such as foam or honeycomb, can enhance the durability and mechanical performance of the cross-arms [1,2].

Developing a reliable and effective model (Equations (3) and (5)) for pGFRP composite cross-arms involves two stages: in-time and out-of-time validation. These steps ensure that the model accurately captures the behaviour of the cross-arm and can be trusted for future predictions. Figure 1 briefly outlines the process, which includes meticulous data gathering, parameter calibration, and validation for model creation (Findley’s power law and the Burger model). Figure 1a depicts out-of-time validation, which evaluates the model’s performance using the complete assembly of a pGFRP cross-arm to determine the maximum deformation [44]. The technique of assessing a model’s performance using data from the same timeframe as Figure 1a is known as in-time validation, according to Figure 1b [44,45]. By contrasting the predictions with the actual data collected throughout that time, this stage confirms the model’s accuracy in detecting patterns and relationships. This evaluates the model’s capacity for generalisation and prediction accuracy using fresh, untested data. Both in-time and out-of-time validation are crucial for a model development to be trustworthy and effective since they show how the model operates in various circumstances or times. Additionally, Section 2 below provides further detail on the flexural creep analysis utilising the Findley’s power law and Burger models shown in Figure 1.

Understanding the creep analysis and durability of a material requires a comprehensive understanding of its behaviour under constant stress and strain, considering factors such as viscosity and environmental conditions [17,46]. This study on the reduction factor of pGFRP composite cross-arms aims to investigate the long-term performance and durability of these materials in overhead power transmission lines. Mathematical modelling and analysis of mechanical properties were employed to predict the reduction factor over time. The study involved manufacturing three types of pGFRP composite cross-arms, subjecting them to various mechanical tests, and developing a mathematical model that considered material properties, environmental conditions, and loading conditions.

While previous research has addressed certain limitations, such as data limitations, material variability, environmental factors, and cost-effectiveness, this study shows that simple mathematical modelling and analysis can be used to forecast the long-term performance and durability of composite materials in overhead power transmission lines. The results also provide insights into environmental factors influencing the reduction factor of pGFRP cross-arms, enabling better design and maintenance practices for these structures.

## 2. Materials and Methodology

### 2.1. Pultrusion Glass Fibre-Reinforced Polymer (GFRP) Cross-Arm Tube

The GFRP members constituting the cross-arm in transmission towers are manufactured using a technique called pultrusion. Pultrusion is a manufacturing process typically used to produce long members of composite materials with constant cross-sections; hence, these members usually have hollow square or rectangular cross-sections. Cross-arms with pGFRP members have high-strength fibreglass mats and strands of reinforced fibreglass which give them stiffness and strength in multiple directions. These fibreglass strands and mats are initially submerged in a liquid resin and then pulled through a hot steel forming die which results in a robust and stiff fibreglass-reinforced composite hollow member. The arranging of various layers of materials to produce a composite structure typically involves the schematic layering of a pGFRP (pultruded glass fibre-reinforced polymer) specimen (Figure 2). Nevertheless, depending on the precise design needs, the layering configuration can change. It is vital to remember that the intended use, design considerations, and engineering needs can all change the specific stacking sequence, the number of layers, and the fibre orientation. To maximise the performance of the material in terms of strength, stiffness, and other desired attributes, the stacking pattern is carefully chosen.

The specimens for this investigation came from various manufacturers of composite cross-arms. The study attempted to capture the inherent variability in material qualities, fabrication processes, and product quality across the industry by integrating specimens with various production sources. Furthermore, using specimens from different manufacturers allows for a more robust evaluation of the reduction factor and life-span prediction models. It is helpful in considering the possibility of material performance and behaviour variances due to differences in production methods, input materials, and quality assurance procedures. This approach improves the reliability and applicability of the mathematical models built for life-span prediction by accounting for the industry’s potential fluctuations and uncertainties and thereby offer valuable insights and enables early predictions for real-world applications of pGFRP composite cross-arm, enhancing their practicality and effectiveness.

By considering the fibre volume fraction as a distinguishing factor, the study ensures that any variations in mechanical properties and behaviour can be attributed to the different specimen compositions. Therefore, to determine the ultimate flexural strength and the creep behaviour, three replicate coupons were prepared for each type of fibre volume fraction to ensure that any variations in mechanical properties and behaviour could be attributed to the different specimen compositions. Three pGFRP specimens were cut into each coupon sample. These specimens with dimensions of 38 mm × 380 mm were cut and tested in static failure four-point flexural tests (ASTM D672), tensile tests (ASTM 3039), and flexural creep tests (ASTM D6272) with three different load levels. The material properties of the pGFRPs used in this study are reported in the following section.

### 2.2. Fibre Volume Fraction and Density

The study identified different specimens (polyester with E-Glass fibres) based on their fibre volume fraction. They were labelled PS-1, PS-2, and PS-3, with fibre volume fractions of 61.95%, 67.40%, and 60.85%, respectively. This categorization using fibre volume fraction provides important information about the composition and characteristics of each specimen. Diversity, strength, and behaviour are altered by the specific design of the various layers in the pultrusion process at different interlaminar degrees of glass fibre incorporation. Therefore, to determine the fibre volume fraction, the pGFRP specimens were cut into sizes of 20 mm by 20 mm and subjected to burn-off tests by ASTM D2584. The specimens were burned to 600 °C in an electric oven (furnace) for one hour to completely burn the resin or other materials. The remaining glass fibres were then carefully separated and weighed, yielding the percentage of leftover fibres.

Meanwhile, the displacement method (ASTM D792-00) was used to determine the density and specific gravity (relative density) of a composite material using a densometer. The pGFRP specimens were weighed after being cut into a size of 40 mm by 15 mm. The specimen was then carefully lowered into the distilled water in the densometer, a specific apparatus, which had been filled with water. The apparent mass of the specimen in the liquid was measured by the densometer. The density and specific gravity of the composite material were determined by comparing the apparent mass of the liquid to that of the specimen in the air. The data from this material characterization offered useful insights into the properties of the pGFRP composites, which are shown in Table 2.

### 2.3. Mechanical Test Composite Cross-Arm

#### 2.3.1. Quasi-Static Tensile Test

Tensile tests were conducted on the three types of pGFRP specimens designated as PS-1, PS-2, and PS-3 using a Zwick Roell, Z100, 150 kN Universal Testing Machine (UTM) at the Material Laboratory, Universiti Tenaga Nasional (UNITEN). Each specimen was tested with a loading rate of 5 mm/min, and the test was repeated with three replicates of each specimen to obtain an average value based on ASTM D3039, as shown in Figure 3, while the maximum tensile loads of each specimen and type of cross-arm are presented in Table 3. Additionally, the tensile setup of this machine has anchors at the two ends of the tensile specimens to ensure proper fixation, uniform distribution of the applied load, and to prevent slippage or detachment during the application of the tensile load, which will result in inaccurate measurement of tensile strength and elongation.

Based on Table 3, PS-3 sample had the highest tensile load, which was 73.57 kN, whereas PS-1 had the lowest tensile load of 61.907 kN. This observation was due to the PS-3 specimen having a higher fibre content compared with PS-1. The increase in fibre content in the polymer composite would allow better tensile strength and stiffness due to better stress transfer within the matrix.

#### 2.3.2. Quasi-Static Flexural 4-Points Bending Test

The ultimate strength flexural test (4-points bending) of cross-arm specimens tested using a UTM machine by Zwick/Roel Z100, Zwick/Roel, Ulm, Germany (Figure 4), and the data obtained are reported in Table 4.

The parameters of the polymer E-Glass UD (unidirectional) in Table 4 were utilised to replicate the numerical analysis for the aforesaid specimens to forecast the flexural deflection and cross-arm life-span. Furthermore, load versus mid-span deflection was recorded to determine stresses and strains using elastic beam theory [47], as shown below in Equations (1) and (2):(1)$$\sigma =\frac{3P\left(L-L_{i}\right)}{2{\mathrm{bd}}^{2}}$$
(2)$$\varepsilon =\frac{6\left(L-L_{i}\right)d\Delta }{4a^{3}-3{\mathrm{aL}}^{2}}$$
where σ is stress in the outer fibre in (MPa), P is the load in (N), L is the support span in (mm), L_i_ is the loading span in (mm), b is the specimen width (mm), d is the specimen thickness (mm), ∆ is the midspan deflection (mm) and a is the distance from the support to the nearest loading point (mm).

#### 2.3.3. Flexural Creep Test

The pGFRP cross-arm specimens were subjected to a flexural creep test over 30 days (720 h) in the Civil Engineering Laboratory at UNITEN, where the average temperature was 25.7 ℃ and the relative humidity was 80.35%. Figure 5a depicts the experimental arrangement used to evaluate the behaviour of various types of composite cross-arm specimens subjected to specific flexural creep loads (12%, 24%, and 37%) in the laboratory flexural creep test. Deflection measurements were taken at the mid-span of the coupon samples throughout a period (720 h), and the deflection was measured every 15 min (daily for an hour of monitoring). In this experiment, the deflection was tracked using dial gauge measurements, as shown in Figure 5b. The schematic diagram of the test can be observed in Figure 5b, and the four-point bending creep test setup is conducted according to ASTM D6272. The purpose of using four-point bending is to ensure that the sample is subjected to constant flexural stress at the sample midpoint.

### 2.4. Stiffness Reduction over Time Analysis

An engineering technique called stiffness reduction over time analysis is applied to forecast how structures will behave over the long term under various loading scenarios. The examination evaluates how a material loses stiffness over time because of numerous elements such as corrosion, fatigue, and creep. This is significant because long-life constructions must endure loads and environmental factors over a long period of time. Usually, there are several steps to the study of how stiffness reduces over time.

The degradation model is used to predict the reduction in the stiffness of the material over time, enabling engineers to evaluate the structure’s long-term performance for various loading scenarios and environmental variables. It can be performed using power-law models for creep, fatigue crack growth models for fatigue, and corrosion models for corrosion. Finally, the stiffness reduction results can help the engineer optimise the structure’s design. Moreover, the stiffness reduction study might lead to revising the material selection, shape, or load conditions to ensure that the structure meets its performance criteria during its predicted lifetime. Therefore, stiffness reduction over time analysis is a crucial tool for estimating the long-term performance and safety of structures in various applications, including bridges, buildings, aircraft, and spacecraft.

#### 2.4.1. Findley’s Power Law Model Flexural Creep Analysis

To evaluate the creep properties of the pGFRP composite cross-arms, it is crucial to have preliminary knowledge of its flexural bending characteristics. This initial information serves as a foundation for the subsequent analysis of creep behaviour, such as empirical and physical models. A widely known empirical model is the Findley power law model, which is implemented to evaluate the creep strain within a certain period in accordance with the stress factor and material constants [19]. The model aids in the analysis of creep behaviour of composites by removing exaggerated data deviation since it is simple and straightforward [18]. The mathematical formula of this model [48] is presented as Equation (1):(3)$$\varepsilon_{t}={\mathrm{mt}}^{n}+\varepsilon_{0}$$
where m and n refer to the load constant and specific material exponent, respectively, and are obtained from experimental data curve fittings, and $\varepsilon_{0}$ is the instantaneous strain after the load is applied. By substituting the elastic strain with applied stress σ divided by the modulus of elasticity E_o_, Equation (2) is obtained.
(4)$$\varepsilon \left(\sigma ,t\right)=\frac{\sigma }{E_{o}}+{m*t}^{n}$$

This power law equation is used to simulate the creep behaviour for GFRP laminates and predict the strain at a specified time (t) and stress (σ), after obtaining the material-specific parameters m and n from curve fittings. Moreover, Findley’s model can correctly capture the primary and secondary stages of creep for FRP materials while being simple and easy to implement. However, this model is typically used at temperatures around 25 °C, i.e., room temperature, while other models, such as the Burger model, are utilized for higher temperatures.

#### 2.4.2. Burger Model Flexural Creep Analysis

As a physical-type model, the Burger model consists of mathematical equations which represent the creep trends using a spring and dashpot diagram [49]. Most studies utilize the Burger model to evaluate viscoelasticity behaviour of material under creep load [25,50]. For instance, Perez et al. [51] and Chandra and Sobral [52] established that creep strain comprises three main aspects: viscoelastic strain (Kelvin’s dashpot element); instantaneous strain (Maxwell spring); and viscous strain (Kevin–Voight element). Figure 6 shows the Burger model, which includes the three major elements, which the model can represent as Equation (3).
(5)$$\varepsilon_{t}=\varepsilon_{e}+\varepsilon_{d}+\varepsilon_{v}$$

Equation (5) includes ${\;\varepsilon }_{e}$, $\varepsilon_{d}$ and $\varepsilon_{v}$, which refer to the elastic strain, viscoelastic strain, and viscous strain, respectively. In specific terms, Equation (6) is derived based on Equation (5) and the physical elements of Burger models, such as spring and dashpot elements.
(6)$$\varepsilon \left(t\right)=A+B [1-e^{-\mathrm{Ct}}]+\mathrm{Dt}$$
where A is a parameter that represents the instantaneous elastic strain $\varepsilon_{0}$, while B, C, and D are variables that correspond to the viscosity parameters which are represented as the dashpots and the elastic moduli of the springs for this model, and t is the time in hours [54]. These parameters can be determined based on data fitting from experimental data and functions to characterise the creep behaviours of composites. In general, the first term of the Burger model aids the analysis by finding the constant independent of time, whereas the second term elaborates on the early stage of creep but reaches a maximum quickly. The third term represents the long-term creep trend at a constant creep rate. By substituting the elastic strain A with the applied stress σ divided by the modulus of elasticity E_o_, we obtain Equation (7).
(7)$$\varepsilon \left(\sigma ,t\right)=\frac{\sigma }{E_{o}}+B [1-e^{-\mathrm{Ct}}]+\mathrm{Dt}$$

After determining the coefficients B, C, and D from experimental data curve fittings, this equation allows engineers to simulate the creep behaviour of GFRP laminates and predict the strain at a specified time (t) and stress (σ). Furthermore, the Burger model can capture the instant strain, the early stage of creep, and the long-term creep at a constant creep rate with its three components.

## 3. Results and Discussion

### 3.1. Creep Strain Failure—Reduction Factor, $\chi \left(t\right)$

The reduction factor is a metric that is often utilised in the recommendations to manage the elastic resistance of a structure. To evaluate the lateral load resistance system in this study (through inelastic behaviour), the samples (coupon and full-scale specimens) were subjected to creep with three different loading levels: 12%, 24%, and 37%. Therefore, the theory of Findley’s power law model and Burger model are used to find results for the flexural stiffness reduction value for a pGFRP composite cross-arm due to decreases in elastic modulus (strain) and in parallel with the period of service.

### 3.2. Reduction Factor, $\chi \left(t\right)$—Findley’s Power Law Model Analysis

Findley’s power law model, which was first developed for thermoplastics, can provide a good approximation of the creep effects or viscoelastic properties of the polymer composite. Generally, the Findley power law model is used to characterise the time-dependent behaviour and stiffness reduction of materials, especially under cyclical loading conditions. Therefore, the reduction factor for GFRP can be analysed using Equation (8).
(8)$$\chi \left(t\right)={\left(1+\frac{E_{0}}{E_{t}}t^{n}\right)}^{-1}$$

The reduction factor χ(t) in this model is defined as a function of the cumulative plastic strain and the fatigue stiffness of the material which is quantified as the average value of E_o_, n, and m in Table 5. Meanwhile, the stiffness loss over time for composite cross-arms subjected to prolonged stress or creep can be complex and significant. By multiplying the time-dependent reduction factor χ(t) with the Findley power law factor, both the time-dependent behaviour and the accumulated plastic strain-related stiffness reduction in the material can be determined. This combined approach allows consideration of both the time-dependent effects and the cyclic loading-induced stiffness reduction. In the Findley power law model, the time-dependent reduction factor χ(t), is used to modify the elastic modulus of the material over time. The modified elasticity modulus E(t) is given by Equation (9):(9)$$E\left(t\right)=E_{0}*\chi \left(t\right)$$

Then, the function χ(t) can be used to estimate the decrease in stiffness over time for each loading level, and the structure’s stability and safety can be evaluated accordingly. By using the Findley power law model and combining the elasticity modulus E(t) with the time-dependent reduction factor χ(t), the changes in mechanical properties of a material over time due to both the effects of constant stress and time can be estimated. The results of the χ(t) and $E\left(t\right)$ functions are indicated in Table 5.

The reduction factor χ(t) is a measure of how much a material’s stiffness has decreased compared with its starting stiffness. The material has largely maintained its initial stiffness when the reduction factor χ(t) is close to 1, and the reduction in strain is small. On the other hand, when the reduction factor χ(t) approaches 0, it denotes a severe loss of stiffness, causing a greater loss of strain. This value indicates how much the material’s distortion has decreased from its initial deformation. It is crucial to understand that the reduction factor (t) can be determined using empirical data or mathematical models tailored to the material and loading circumstances. Therefore, the accuracy of the estimation of strain reduction depends on the dependability of the reduction factor χ(t) incorporated in the calculation.

The data of strain reduction (percent) and reduction factor (t) can be analysed using curve fitting techniques. Regression analysis, commonly referred to as curve fitting, is a mathematical method for identifying the curve or function that best fits a given set of data points [55]. Finding a mathematical equation or model that roughly represents the relationship between the independent and dependent variables in the data is required. Curve fitting aims to decrease the discrepancy between the fitted curve and the actual data points, enabling parameter estimation and value prediction within or even outside the data range. The nature of the data and the anticipated relationship between the variables influence the fitting method selection. Numerous fields, including physics, engineering, finance, biology, and the social sciences, use curve fitting extensively [56,57]. To compute the strain reduction using the reduction factor χ(t), the strain at time t is compared with the original strain. The method described in the following section can be used to determine the percentage reduction in strain, as shown in Table 6 below.

### 3.3. Reduction Factor, $\chi \left(t\right)$—Burger Model Analysis

The function $\varepsilon \left(t\right)$ from Equation (7) can be changed to account for the time-dependent reduction factor resulting from creep in the case of composite materials subjected to various loading levels. The Burger model describes the total strain with three components, an elastic time-independent component, a viscoelastic time-dependent component, and a viscous time-dependent component, which are indicated in Equation (10).
(10)$$\chi \left(t\right)={\left(1+\frac{E_{0}}{E_{b}} [1-e^{-\mathrm{Ct}}]+\frac{E_{0}}{E_{d}}t\right)}^{-1}$$

This equation gives the modified elasticity modulus E(t) at a specific time t. It considers the initial elasticity modulus $E_{0}$ and modifies it based on the reduction factor χ(t), which considers the effects of creep and damage accumulation. By multiplying the initial elasticity modulus $E_{0}$ by the reduction factor χ(t), the modified elasticity modulus E(t) is obtained. This modification accounts for the time-dependent reduction in strength and stiffness due to creep and damage accumulation, as captured by the reduction factor χ(t).

It is important to note that the specific values of $E_{b}$, C, and $E_{d}$ should be determined based on experimental data or established from correlations in the literature for the specific material and loading conditions. The modified elasticity modulus E(t) in Equation (11) provides a representation of the evolving mechanical properties of the material over time, incorporating the effects of creep and damage.
(11)$$E\left(t\right)=E_{0}*\chi \left(t\right)$$

Furthermore, the Burger model can capture the instant strain, the early stage of creep, and the long-term creep at a constant creep rate with data for its three components (B, C and D) from Equation (6), as shown in Table 7. The Burger model parameters for pGFRP specimens (PS-1, PS-2, and PS-3) under various loading circumstances are also shown in Table 7. The parameters are the percentage of loading, stress (σ) in MPa, and $E_{0}$ = σ/$\varepsilon_{0}$ in GPa. These variables enable precise modelling and comprehension of the pGFRP specimens’ reaction to various loading circumstances and offer useful insights into the mechanical behaviour and characteristics of the materials.

Table 8 presents the data for the pGFRP specimens (PS-1, PS-2, and PS-3) after 720 h. Different amounts of stress were applied to the specimens, and the behaviour was then examined. The determined parameters included the applied loads, computed stresses, thickness, strain values, coefficients, effective modulus ($E_{b}$), damage modulus ($E_{d}$), initial modulus ($E_{0}$), reduction factor (t), and modulus values at 720 h E(t). These results show how the modulus values of the pGFRP specimens change over time and under different stress levels, shedding light on their mechanical characteristics and deterioration.

The data shown in Table 9 depict how three different pGFRP cross-arm types (PS-1, PS-2, and PS-3) behaved for 18,250 months. All cross-arms incur an increase in deflection and strain with time. For PS-1, the maximum equivalent strain rose from 0.35% to 8.15%, while the total deflection rose from 35.68 mm to 829.61 mm. The maximum equivalent strain on the PS-2 cross-arm increased from 0.18% to 7.26%, and the total deflection increased from 18.37 mm to 744.78 mm. The maximum equivalent strain for the PS-3 cross-arm also increased from 0.24% to 7.75%, while the total deflection increased from 20.62 mm to 713.33 mm. These results illustrate the effect of long-term loading on the structural behaviour of the pGFRP cross-arms by showing a progressive increase in deflection and strain over time. Therefore, the reduction factor through the Burger model is more suitable for the prediction of failure criteria for creep deflection of GFRP specimens in general.

### 3.4. Life Span Prediction—Ultimate Strain Limit

Through these creep strain limit criteria for future life-span prediction, the general equations are obtained for the Findley power law model and the Burger model. Equations (3) and (5) were calculated and plotted in Figure 7 and Figure 8, respectively. The two models diverge significantly where the Burger model continues increasing steadily, while Findley’s model plateaus without any indication of failure. Otherwise there was very little change after the first phase of strain behaviour. For the life-span prediction of all specimens with Burger model failure, the strain criterion reaches the ultimate strain limit (Figure 8).

Considering the predictions are based on a creep test conducted in a room temperature environment, the Burger model estimates for pGFRP cross-arm composite failure over a 10-year period of life-span may be more reasonable. Therefore, the four-element Burger body has many advantages for the prediction of creep behaviour due to the insertion of the Kelvin body between the spring and dash spot of the Maxwell body. In addition, a Findley power law was chosen to model the short-term creep performance of this material because of its past success as an effective modelling tool.

## 4. Conclusions

This study investigated the reduction factor of pultruded glass fibre-reinforced polyester composite cross-arms and developed mathematical models (Burger and Findley power laws) to estimate their life-span. The reduction factor values obtained from these models provide crucial information about the structural behaviour and durability of the composite cross-arms, enabling predictions of their remaining strength and load-bearing capacity over time. The findings offer valuable insights into anticipating the life-span of these cross-arms, assisting engineers and industry experts in determining the expected functional life-span of these structural components. Considering the reduction factor values when making decisions about maintenance and replacement can significantly enhance the safety and reliability of power transmission systems. This study contributes to the existing knowledge on reduction factor analysis and life-span estimation of pultruded glass fibre-reinforced polyester composite cross-arms. Furthermore, the comparison of mathematical models highlights the importance of selecting an appropriate model, and the computed reduction factor values facilitate estimating the remaining service life of the composite cross-arms. Future research efforts should focus on improving the mathematical models and incorporating additional factors to enhance the precision and applicability of life-span estimates. Overall, this study provides valuable insights into the reduction factor and life-span estimation of these composite cross-arms, emphasizing the significance of model selection and the practical implications for determining their remaining service life.

## Figures and Tables

**Figure 1 materials-16-05328-f001:**
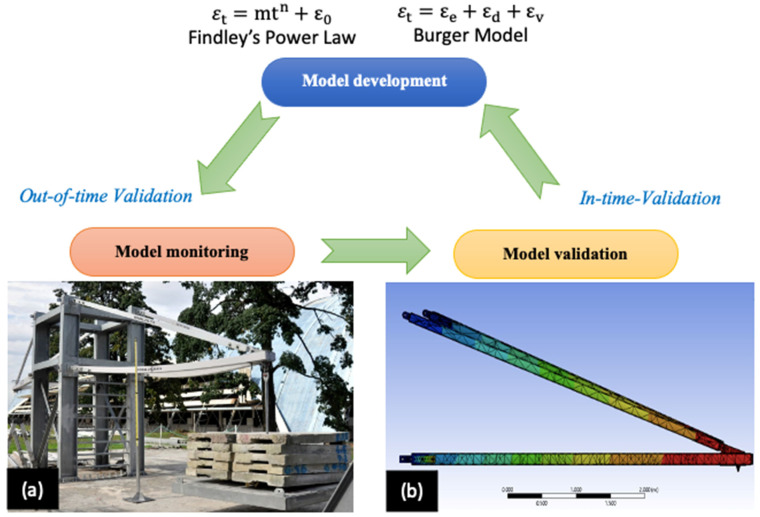
Two main stages of the model development process (Equations (3) and (5)) used to assess the accuracy and reliability of a model structure pGFRP composite cross-arm: (**a**) model monitoring [44], and (**b**) model validation [44,45]. In contrast, the Burger model consists of ${\;\varepsilon }_{e}$, $\varepsilon_{d}$ and $\varepsilon_{v}$, which are classified as the elastic strain, viscoelastic strain, and viscous strain, respectively. The load constants and specific material exponents m and n for the Findley’s power law’s total strain are derived using curve fits of experimental data, while ε_o_ refers to the instantaneous strain.

**Figure 2 materials-16-05328-f002:**
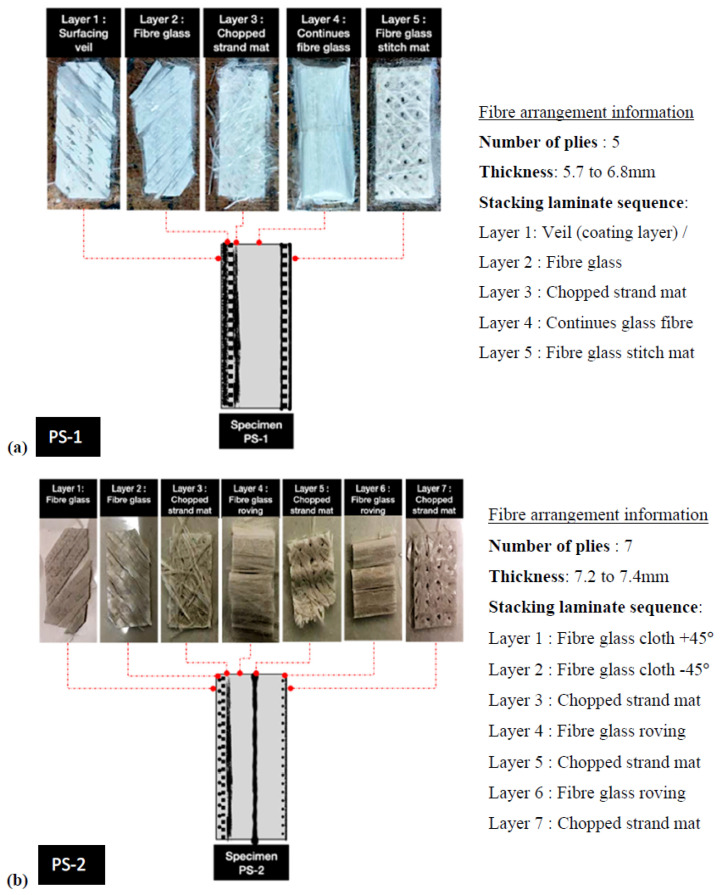
Fibre arrangement and laminate profile of pGFRP composite cross-arm specimens: (**a**) PS-1, (**b**) PS-2, and (**c**) PS-3.

**Figure 3 materials-16-05328-f003:**
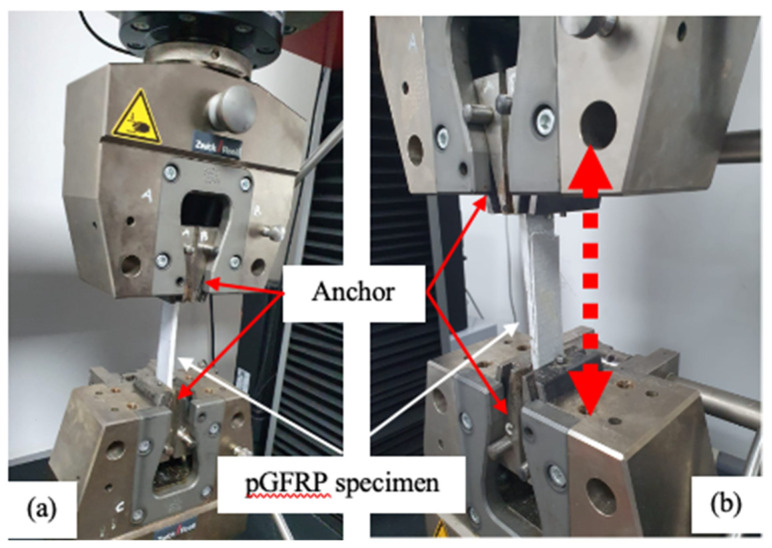
Tensile test of a pGFRP composite cross-arm specimen: (**a**) setup and (**b**) physical change after test.

**Figure 4 materials-16-05328-f004:**
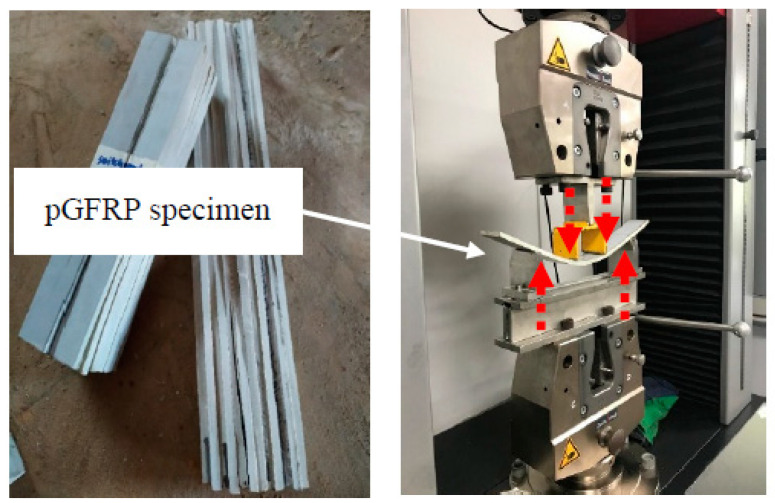
Flexural test of a pGFRP composite cross-arm (4-point flexural bending) performed on a Zwick/Roel Z100 UTM machine.

**Figure 5 materials-16-05328-f005:**
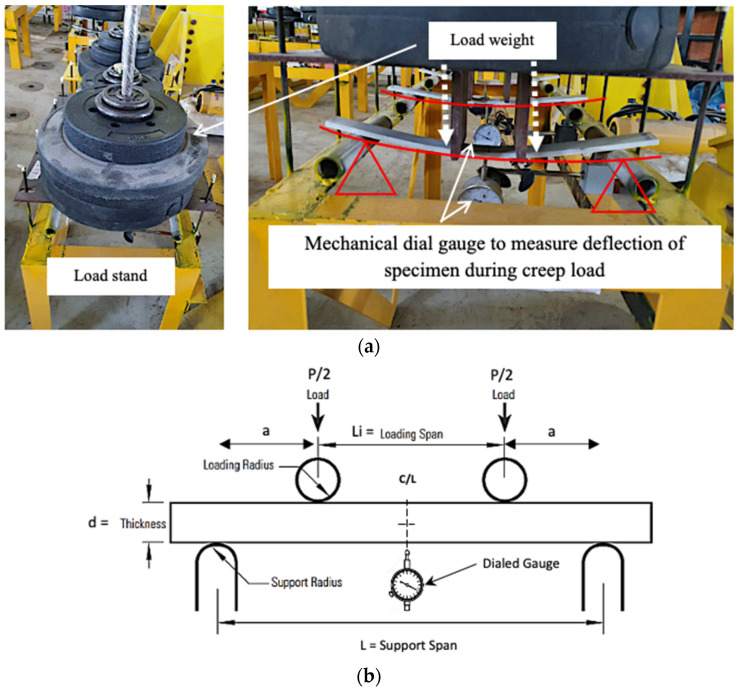
(**a**) Indoor flexural creep test for pGFRP composite specimens with 12%, 24% and 37% load levels. (**b**) Schematic diagram of flexural creep test as per ASTM D6272.

**Figure 6 materials-16-05328-f006:**
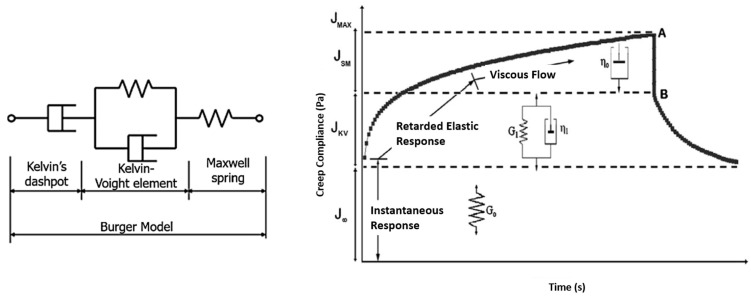
Schematic diagram of physical Burger model [53].

**Figure 7 materials-16-05328-f007:**
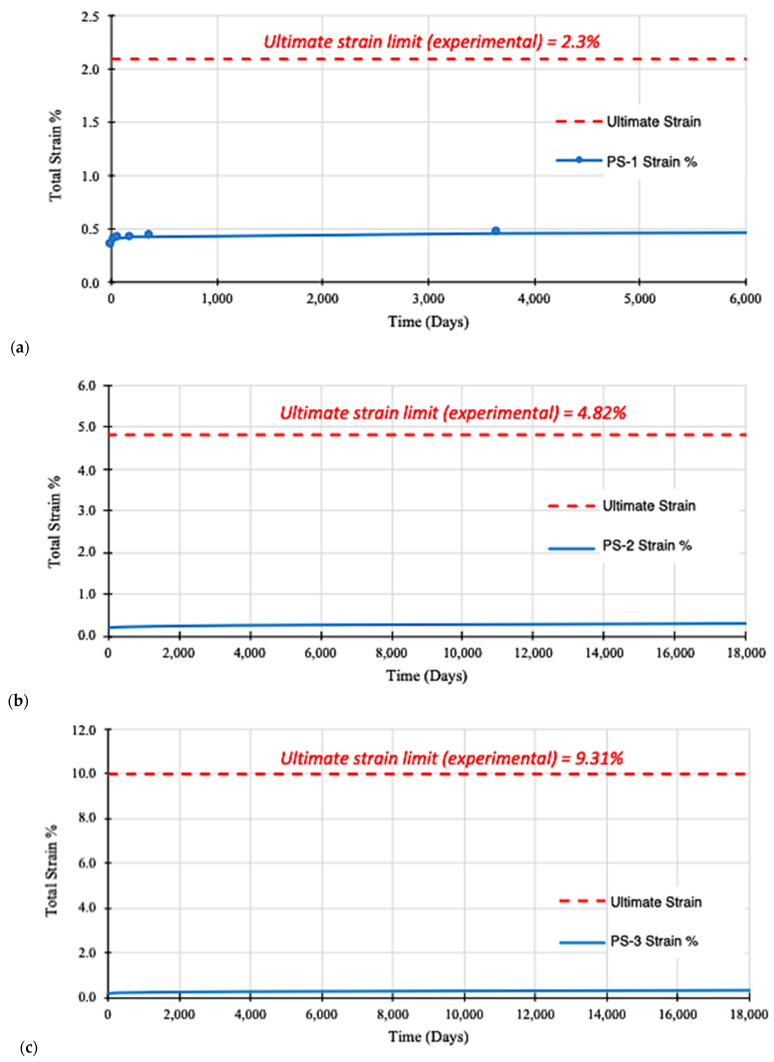
Findley’s power law model analysis for life-span prediction of pGFRP specimens: (**a**) PS-1, (**b**) PS-2, and (**c**) PS-3.

**Figure 8 materials-16-05328-f008:**
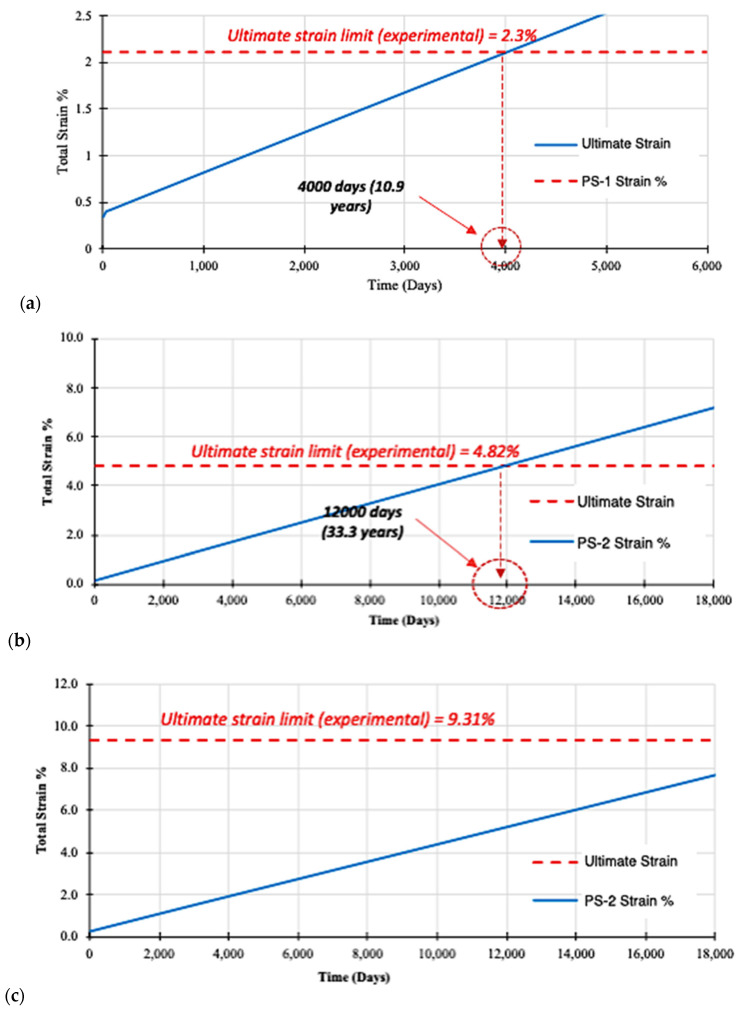
Burger model analysis for life-span prediction of pGFRP specimens: (**a**) PS-1, (**b**) PS-2, and (**c**) PS-3.

**Table 1 materials-16-05328-t001:** Creep analysis study of pultruded GFRP composites.

Material	Type of Specimen	Type of Loading	Testing Duration	References
GFRP/Vinyl ester	Frame	Flexural (3/4-points)	3500–10,000 h	[29]
GFRP/Polyester	Beam assembly		24 h	[30]
	Sheet piling		9000 h	[31]
	Coupon		720–1000 h	[19,32]
	Section profile		270–1600 h	[33]
GFRP/Vinyl ester	Prismatic	Compression	2500–10,000 h	[34,35,36]
GFRP/Polyester	Columns			

**Table 2 materials-16-05328-t002:** Material properties of pGFRP.

Properties	pGFRP Specimens		
	PS-1	PS-2	PS-3
Density (kg/m^3^)	1.83	1.85	1.87
Fibre volume fraction, $V_{f}$ (%)	61.95	67.40	60.85

**Table 3 materials-16-05328-t003:** Tensile load of various pGFRP composite cross-arm specimens.

No.	Specimens	Mechanical Properties	
		Max. Tensile Force (kN)	Load (kgf)
1	PS-1	61.91	6310.60
2	PS-2	29.53	3009.79
3	PS-3	73.57	7499.08

**Table 4 materials-16-05328-t004:** The mechanical properties of various pGFRP composite cross-arm specimens were tested using the Zwick/Roel Z100 UTM machine.

No.	Specimen	Mechanical Properties	
		Max. Flexural Force (N)	Load (kgf)
1	PS-1	972.6	99.1
2	PS-2	1207.15	123
3	PS-3	759.06	77.4

**Table 5 materials-16-05328-t005:** Results of the reduction factor, χ(t), for pGFRP specimens.

Parameter	PS-1			PS-2			PS-3		
	Stress Level			Stress Level			Stress Level		
	12%	24%	37%	12%	24%	37%	12%	24%	37%
Thickness d (mm)	6	6	6	7.2	7.2	7.2	5.2	5.2	5.2
Applied Load (N)	116.4	233.0	360.5	144.8	289.7	446.6	91.1	182.2	280.9
Stress σ (MPa)	29.87	59.78	92.50	25.80	51.62	79.58	31.12	62.24	95.96
$\varepsilon_{0}$ (10^−3^)	0.0026	0.0039	0.0055	0.0042	0.0046	0.0073	0.0045	0.0050	0.0075
n	0.1481	0.3461	0.1481	0.1586	0.3238	0.3446	0.343	0.1099	0.1722
m (ε %)	0.0301	0.0055	0.0301	0.0309	0.0057	0.0095	0.0044	0.0451	0.0292
E_0_ (GPa)	11.6	15.5	17.0	6.1	11.2	10.9	6.99	12.4	12.8
E_t_ = σ/m (GPa)	99.2	1086.9	307.3	83.5	905.5	837.6	707.3	138.0	328.6
t (hours)	720			720			720		
Reduction Factor χ(t)	0.89			0.91			0.90		
E(t) (GPa)	13.12			8.6			9.69		

**Table 6 materials-16-05328-t006:** Stiffness reduction in strain pGFRP cross-arms specimens.

Reduction (Strain),%		0%	14%	15%	17%	19%	25%	29%
PS-1	Total Deflection (mm)	35.7	41.3	41.9	43.0	43.8	47.0	50.0
	Max Equivalent Strain (%)	0.35	0.41	0.41	0.42	0.43	0.46	0.49
	Time (Days)	0	30	60	180	365	3650	18,250
	Time (Months)	0	1	2	6	12	120	608.3
**Reduction (strain),%**		**0%**	**11%**	**13%**	**15%**	**16%**	**23%**	**28%**
PS-2	Total Deflection (mm)	18.37	20.17	20.55	21.32	21.96	25.13	28.90
	Max Equivalent Strain (%)	0.18	0.20	0.20	0.21	0.22	0.25	0.29
	Time (Days)	0	30	60	180	365	3650	18,250
	Time (Months)	0	1	2	6	12	120	608.3
**Reduction (strain),%**		**0%**	**12%**	**15%**	**20%**	**24%**	**40%**	**53%**
PS-3	Total Deflection (mm)	20.62	22.59	22.89	23.48	23.93	25.97	28.10
	Max Equivalent Strain (%)	0.24	0.26	0.26	0.27	0.27	0.30	0.32
	Time (Days)	0	30	60	180	365	3650	18,250
	Time (Months)	0	1	2	6	12	120	608.3

**Table 7 materials-16-05328-t007:** Determination of Burger model parameters for pGFRP cross-arm specimens.

**PS-1**	**Loading (%)**	**Stress σ, (MPa)**	**A = ** $\varepsilon_{0}$ ** (10^−3^)**	**B** **(10^−4^)**	**C** **(10^−3^)**	**D** **(10^−7^)**	$E_{o}$ ** = σ/ε_o_ (GPa)**
	*12*	46.85	2.57	4.78	761.51	5.32	18.25
	*24*	59.78	3.85	2.12	257.19	3.05	15.52
	*37*	92.50	5.46	4.78	760.31	5.31	16.91
	**Average**			**3.90**	**593.00**	**4.56**	**16.89**
**PS-2**	**Loading (%)**	**Stress σ, (MPa)**	**A = ** $\varepsilon_{0}$ ** (10^−3^)**	**B** **(10^−4^)**	**C** **(10^−3^)**	**D** **(10^−7^)**	$E_{o}$ ** = σ/ε_o_ (GPa)**
	*12*	37.43	4.24	5.54	612.34	4.57	8.83
	*24*	63.30	4.62	2.10	264.51	3.39	13.69
	*37*	91.62	7.32	3.41	279.36	8.31	12.52
	**Average**			**3.68**	**385.40**	**5.42**	**11.68**
**PS-3**	**Loading (%)**	**Stress σ, (MPa)**	**A = ** $\varepsilon_{0}$ ** (10^−3^)**	**B** **(10^−4^)**	**C** **(10^−3^)**	**D** **(10^−7^)**	$E_{o}$ ** = σ/ε_o_ (GPa)**
	*12*	53.16	4.45	1.32	384.95	4.68	11.94
	*24*	84.16	5.01	6.38	1002.82	4.78	16.81
	*37*	117.84	7.51	4.88	500.67	4.29	15.69
	**Average**			**4.20**	**629.48**	**4.58**	**14.81**

**Table 8 materials-16-05328-t008:** Determination of reduction factor χ(t) and data analysis of pGFRP specimens (t = 720 hrs).

Parameter	PS-1			PS-2			PS-3		
	Stress Level, σ			Stress Level, σ			Stress Level, σ		
	12%	24%	37%	12%	24%	37%	12%	24%	37%
Thickness (d), mm	6	6	6	7.2	7.2	7.2	5.2	5.2	5.2
Applied Load (N)	116.4	233.0	360.5	144.8	289.7	446.6	91.1	182.2	280.9
Stress (σ), Mpa	29.87	59.78	92.50	25.80	51.62	79.58	31.12	62.24	95.96
A = ε_0_	0.003	0.004	0.006	0.004	0.005	0.007	0.0045	0.0050	0.0075
B (10^−4^)	4.78	2.12	4.78	5.54	2.10	3.41	1.32	6.38	4.88
C (10^−3^)	761.51	257.19	760.31	612.34	264.51	279.36	384.95	1002.8	500.67
D (10^−7^)	5.32	3.05	5.31	4.57	3.39	8.31	4.68	4.78	4.29
E_b_ = σ/B (10^3^)	62.4	282.3	193.3	46.6	246.4	233.4	236.5	97.5	196.3
E_d_ = σ/D (10^5^)	561.6	1957.2	1741.1	564.7	1523.0	957.6	664.6	1301.4	2237.6
E_0_ (Gpa)	11.6	15.5	17.0	6.1	11.2	10.9	6.99	12.4	12.8
t (hour’s)	720			720			720		
Reduction Factor χ(t)	0.86			0.89			0.89		
E(t) (Gpa)	12.71			8.37			9.62		

**Table 9 materials-16-05328-t009:** Reduction stiffness (strain) of creep deflection for pGFRP cross-arm specimens.

Reduction (Strain)		0%	27%	29%	37%	47%	85%	96%
PS-1	Total Deflection (mm)	35.68	41.35	42.58	47.79	55.81	198.10	829.61
	Max Equivalent Strain (%)	0.35	0.41	0.42	0.47	0.55	1.96	8.15
	Time (Days)	0	30	60	180	365	3650	18250
	Time (Months)	0	1	2	6	12	120	608.3
**Reduction (strain)**		**0%**	**11%**	**16%**	**31%**	**46%**	**89%**	**98%**
PS-2	Total Deflection (mm)	18.37	20.56	21.75	26.54	33.91	164.63	744.78
	Max Equivalent Strain (%)	0.18	0.19	0.21	0.25	0.33	1.61	7.26
	Time (Days)	0	30	60	180	365	3650	18250
	Time (Months)	0	1	2	6	12	120	608.3
**Reduction (strain)**		**0%**	**10%**	**15%**	**28%**	**43%**	**87%**	**97%**
PS-3	Total Deflection (mm)	20.62	22.91	24.06	28.62	35.65	160.31	713.33
	Max Equivalent Strain (%)	0.24	0.25	0.26	0.31	0.39	1.76	7.75
	Time (Days)	0	30	60	180	365	3650	18250
	Time (Months)	0	1	2	6	12	120	608.3

## Data Availability

Data available on request due to restrictions e.g., privacy or ethical. The data presented in this study are available on request from the corresponding author. The data are not publicly available due to containing information that could compromise the privacy of research participants.

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
