# Peer review of "The Reduction Factor of Pultrude Glass Fibre-Reinforced Polyester Composite Cross-Arm: A Comparative Study on Mathematical Modelling for Life-Span Prediction"

_materials, 2023, doi:10.3390/ma16155328_

Round 1

Reviewer 1 Report

As the Materials journal the topic of the work at hand would appear to be an appropriate one, in particular paying attention to the research.

The article  presents tested materials. The course of the research and methods has been presented correctly.

The abstract is a little bit confuse and missis some information like more conclusions (needs to be systematized and summarized).

 The Introduction section quite briefly refers to the content of the article, of course the authors pay attention to the key theses from the area of literature analysis, but this section should be reduce by a general thematic introduction. 

It would be reasonable for the reader to introduce analysis of the properties of such composites.

most important notes: 

-the research methodology should be described in detail, including the preparation of materials

- fig 1 is illegible

-no detailed description of the preparation of materials - machinery, equipment,  parameters, etc.

- fig 2 is illegible

- How many samples had to be tested in accordance with the standard?, please refer to the standards for testing mechanical properties,

- measurement parameters are not described exactly

- the test results should be analyzed and the reasons for the changes in properties obtained should be indicated,

- the text needs editorial correction in accordance with the requirements of the journal and the arrangement of figures and tables needs to be improved; font, references, etc.

The conclusions do not refer to the work, but to the description of what the work presents. It is recommended to conduct a deeper discussion and refer to the results in the conclusions, also critically presenting the advantages and disadvantages of the method - which does not seem to be difficult when reading the paper.

Minor editing of English language required

Author Response

Manuscript: MDPI: Materials-2435057 (SI: Synthetic and Natural Fiber Reinforced Polymer Matrix Composites for Advanced Applications)

Title: The Reduction Factor of Pultrude Glass Fibre-Reinforced Polyester Composite Cross-Arm: A Comparative Study on Mathematical Modelling for Life-Span Prediction

REVIEWER: 1 

As the Materials journal the topic of the work at hand would appear to be an appropriate one, in particular paying attention to the research. The article  presents tested materials. The course of the research and methods has been presented correctly.

  • The abstract is a little bit confuse and missis some information like more conclusions (needs to be systematized and summarized).

Answer: (Page: Abstract)

We thank the reviewer for the comment. The abstract has been reorganized and now it reflects better what the reviewer suggested.

“This paper presents an experimental and numerical investigation of pultruded composite glass fibre-reinforced polymer (pGFRP) cross-arm subjected to flexural creep behaviour due to the performance and sustainability in composite cross-arm structure application. The primary objective of this study was to investigate the failure creep behaviour of pGFRP cross-arms with different stacking sequences. Specifically, the study aimed to understand the variations in strain rate exhibited during different stages of the creep process. Therefore, this study will emphasize a simplified approach within the experiment, numerical analysis, and mathematical model on three different pGFRP composites to adopt the stiffness reduction factors that will present the prediction of failure. The findings show that Findley's power law and Burger model projected very different strains and diverged noticeably outside the testing period. Findley's model estimated a minimal increase in total strain over 50 years, while Burger model anticipated PS-1 and PS-2 composites would fail within about 11 and 33 years, respectively. Burger Model's forecasts might be more reasonable due to the harsh environment the cross arms are expected to withstand. The endurance and long-term performance of composite materials used in overhead power transmission lines may be predicted mathematically, and this insight into material property factors can help with design and maintenance.”

  • The Introduction section quite briefly refers to the content of the article, of course the authors pay attention to the key theses from the area of literature analysis, but this section should be reduce by a general thematic introduction.

Answer: (Page: 2 - 4)

Thank you for the comment. The authors briefly touch upon the key themes and theses from the literature analysis. By incorporating a general thematic introduction, the readers will have a clearer understanding of the article's focus and relevance within the field. This revision would enhance the coherence and effectiveness of the introduction section, allowing readers to grasp the main points more straightforwardly.

  • It would be reasonable for the reader to introduce analysis of the properties of such composites.

Answer: (Page: 5 -7)

The specimens for this investigation came from various manufacturers of composite cross-arms. The study attempted to capture the inherent variability in material qualities, fabrication processes, and product quality across the industry by integrating specimens with various production sources. Therefore, the text paragraph has been enhanced with an explanation of the preparation of specimen properties.

The specimens for this investigation came from various manufacturers of composite cross-arms. The study attempted to capture the inherent variability in material qualities, fabrication processes, and product quality across the industry by integrating specimens with various production sources. Furthermore, using specimens from different manufacturers allows for a more robust evaluation of the reduction factor and life-span prediction models. It's helpful in considering the possibility of material performance and behaviour variances due to differences in production methods, input materials, and quality assurance procedures. This approach improves the reliability and applicability of the mathematical models built for life-span prediction by accounting for the industry's potential fluctuations and uncertainties and giving more valuable and trustworthy for real-world uses of pGFRP composite cross-arms.

By considering the fibre volume fraction as a distinguishing factor, the study ensures that any variations in mechanical properties and behaviour can be attributed to the different specimen compositions. Therefore, to determine the ultimate flexural strength and the creep behaviour, three (3) replicate coupon were prepared for each type of fibre volume fraction to ensures that any variations in mechanical properties and behaviour can be attributed to the different specimen compositions. Three pGFRP specimen have been cut into coupon sample which having the dimensions of 38 mm x 380 mm were cute and tested in static failure four-point flexural tests ASTM D672, tensile test ASTM 3039, and flexural creep tests ASTM D6272 with three different load levels. The material properties of the pGFRP’s used in this study is reported in the following section.”

most important notes:

  • the research methodology should be described in detail, including the preparation of materials

Answer: (Page: 5 -7)

   Appreciates the reviewer's comment regarding the detail’s description of material – machinery, equipment, parameter, however we able to provide the mechanical properties of each GPRP cross-arm tubes specimens. However, due to confidentiality, author unable to provide specific information about the manufacturer or the manufacturing technology used in this study. It is important to note that the purpose of our research was to investigate the reduction factor and lifespan estimation of the GPRP cross-arm tubes, focusing on their mechanical behaviour rather than the specific manufacturing process.

The author understands the importance of transparency and providing complete information about the materials used in the study. Although we are unable to disclose the manufacturer or product name, we ensured that the GPRP cross-arm tubes used in the experiments were representative of commercially available products and met industry standards. The study results are based on the performance of these tubes and apply to similar commercially available GPRP cross-arm tubes. Again, the author wants to apologise for any confusion the lack of comprehensive manufacturing information may have caused. We intended to maintain confidentiality while providing relevant findings and insights into the reduction factor and lifespan estimation of GPRP cross-arm tubes.

  • fig 1 is illegible

Answer: (Page: 4)

Figure 1 and caption has been improved will reflects better understanding for reader.

  • no detailed description of the preparation of materials - machinery, equipment, parameters, etc.

Answer: (Page: 5 -7)

   Appreciates the reviewer's comment regarding the detail’s description of material – machinery, equipment, parameter, however we able to provide the mechanical properties of each GPRP cross-arm tubes specimens. However, due to confidentiality, author unable to provide specific information about the manufacturer or the manufacturing technology used in this study. It is important to note that the purpose of our research was to investigate the reduction factor and lifespan estimation of the GPRP cross-arm tubes, focusing on their mechanical behaviour rather than the specific manufacturing process.

The author understands the importance of transparency and providing complete information about the materials used in the study. Although we are unable to disclose the manufacturer or product name, we ensured that the GPRP cross-arm tubes used in the experiments were representative of commercially available products and met industry standards. The study results are based on the performance of these tubes and apply to similar commercially available GPRP cross-arm tubes. Again, the author wants to apologise for any confusion the lack of comprehensive manufacturing information may have caused. We intended to maintain confidentiality while providing relevant findings and insights into the reduction factor and lifespan estimation of GPRP cross-arm tubes.

  • fig 2 is illegible

Answer: (Page: 5 - 6)

Figure 2 and caption has been improved will reflects better understanding for reader.

  • How many samples had to be tested in accordance with the standard?, please refer to the standards for testing mechanical properties,

Answer: (Page: 7)

The number of samples that had to be tested following the standards ASTM D672, ASTM 3039, and ASTM D6272 was not specified, but the standards provide guidelines for conducting tensile testing. The number of samples to be tested would typically be determined based on factors such as the study objectives, statistical significance requirements, and the availability of specimens. Otherwise, to determine the ultimate flexural strength and the creep behaviour, three (3) replicate coupons were prepared for each type of fibre volume fraction (PS-1, PS-2, and PS-3) to ensure that any variations in mechanical properties and behaviour can be attributed to the different specimen compositions.

  • measurement parameters are not described exactly

Answer: (Page: 4 - 7)

The text paragraph has been enhanced with an explanation of the measurement parameters of specimend.

Three pGFRP specimen have been cut into coupon sample which having the dimensions of 38 mm x 380 mm were cute and tested in static failure four-point flexural tests ASTM D672, tensile test ASTM 3039, and flexural creep tests ASTM D6272 with three different load levels. The material properties of the pGFRP’s used in this study is reported in the following section.”

“The study identified different specimens (polyester with E-Glass fibres) based on their fibre volume fraction. They were labelled PS-1, PS-2, and PS-3, with fibre volume fractions of 61.95%, 67.40%, and 60.85%, respectively. This categorization using fibre volume fraction provides important information about the composition and characteristics of each specimen. Diversity, strength, and behaviour have been altered by the specific design of the various layered in the pultrusion process at different interlaminar degrees of glass fibre incorporation. Therefore, to determine the fibre volume fraction, the pGFRP specimens were cut into sizes 20 mm by 20 mm and put through burn-off tests by ASTM D2584. The specimens were burned to 600 °C in an electric oven (furnace) for one hour to completely burn the resin or other materials. The remaining glass fibres were then carefully separated and weighed, yielding the percentage of leftover fibres.

Meanwhile, The displacement method of ASTM D792-00 was used to determine the density and specific gravity (relative density) of a composite material using a densometer. The pGFRP speciemens were weighed after being cut into size of 40 mm by 15 mm. The specimen was then carefully lowered into the distilled water in the densometer, a specific apparatus, which had been filled with the water. The apparent mass of the specimen in the liquid was measured by the densometer. The density and specific gravity of the composite material were determined by comparing the apparent mass of the liquid to that of the specimen in the air. Therefore, the data from this material characterization offered useful insights into the properties of the pGFRP composite, which are shown in Table 2.”

  • the test results should be analyzed and the reasons for the changes in properties obtained should be indicated,

Answer: (Page: 4 - 7)

The text paragraph has been enhanced with an explanation of the preparation of machinery, equipment, parameters, etc.

  • the text needs editorial correction in accordance with the requirements of the journal and the arrangement of figures and tables needs to be improved; font, references, etc.

Answer: (Page: Whole manuscript)

The arrangement of figures and tables has been reorganized and now it reflects better what the reviewer suggested.

  • The conclusions do not refer to the work, but to the description of what the work presents. It is recommended to conduct a deeper discussion and refer to the results in the conclusions, also critically presenting the advantages and disadvantages of the method - which does not seem to be difficult when reading the paper.

Answer: (Page: 29)

Appreciate the reviewer's advice. The conclusion has been revised to better represent the reviewer's suggestions.

“This study investigated the reduction factor of pultruded glass fibre-reinforced polyester composite cross-arms and developed mathematical models (Burger and Findley power laws) to estimate their lifespan. The reduction factor values obtained from these models provide crucial information about the structural behaviour and durability of the composite cross-arms, enabling predictions of their remaining strength and load-bearing capacity over time. The findings offer valuable insights into anticipating the lifespan of these cross-arms, assisting engineers and industry experts in determining the expected functional lifespan of these structural components. Taking into account the reduction factor values when making decisions about maintenance and replacement can significantly enhance the safety and reliability of power transmission systems. This study contributes to the existing knowledge on reduction factor analysis and lifespan estimation of pultruded glass fibre-reinforced polyester composite cross-arms. Furthermore, the comparison of mathematical models highlights the importance of selecting an appropriate model, and the computed reduction factor values facilitate estimating the remaining service life of the composite cross-arms. Future research efforts should focus on improving the mathematical models and incorporating additional factors to enhance the precision and applicability of lifespan estimates. Overall, this study provides valuable insights into the reduction factor and lifespan estimation of these composite cross-arms, emphasizing the significance of model selection and the practical implications for determining their remaining service life.”

Reviewer 2 Report

The paper investigates the reduction factor of pultruded glass fiber reinforced polyester composite cross-arm by examining two mathematical models: the Findley power law model and the Burger model. Paper is well organized with nice figure, the research method is reasonable. This study can provide insights into the behavior of pultruded glass fiber reinforced polyester composite cross-arms and offer a valuable comparison of mathematical models for accurate lifespan prediction. However, the authors are encouraged to consider the following comments for necessary improvement.

1.      ABSTRACT:

(1) Please give a brief background at the beginning of the abstract.

(2) Delete the key words of composite and polyester, and rephrase “pultrude glass fibre-reinforced polyester” with “pultrude glass fibre-reinforced polyester composites”.

2.      INTRODUCTION:

(1) During actual service, structures not only bearing loads, but also suffering from the environmental erosion, such as hygrothermal, ultraviolet radiation, freeze-thaw, etc. Authors should introduce the service behavior of composite materials in complex service environments. Maybe the following relevant studies can be reviewed to make necessary supplements in the research background, such as Mechanics of Advanced Materials and Structures, 2023, 30(4):814-834. Composite Structures. 2021, 261: 113285. Engineering Structures, 2023, 274: 115176.”.

(2) Please supplement the missing text in Figure 1.

(3) Compared to previous research results, please clarify the theoretical and methodological innovations of present paper.

3.      MATERIAL AND METHODOLOGY:

(1) Please provide clear images of Figure 2 (a).

(2) Provide the material properties of glass fiber, fibreglass mat, resin.

(3) Loading rate of the testing machine should be given.

(4) Is there an anchor in the two ends of tensile specimens?

(5) Adjust word size to match text (Line 247, 255, 256, 261).

4.      RESULT AND DISCUSSION:

(1) Adjust the letter sizes in all formulas to be consistent.

(2) Is the prediction model discussed in this paper applicable to specimens under the environment and load coupling conditions?

5.      CONCLUSION:

The conclusion should be condensed, please provide the key results and findings, remove the content unrelated information from the conclusion.

It is good. 

Author Response

Manuscript: MDPI: Materials-2435057 (SI: Synthetic and Natural Fiber Reinforced Polymer Matrix Composites for Advanced Applications)

Title: The Reduction Factor of Pultrude Glass Fibre-Reinforced Polyester Composite Cross-Arm: A Comparative Study on Mathematical Modelling for Life-Span Prediction

REVIEWER: 2 

The paper investigates the reduction factor of pultruded glass fiber reinforced polyester composite cross-arm by examining two mathematical models: the Findley power law model and the Burger model. Paper is well organized with nice figure, the research method is reasonable. This study can provide insights into the behavior of pultruded glass fiber reinforced polyester composite cross-arms and offer a valuable comparison of mathematical models for accurate lifespan prediction. However, the authors are encouraged to consider the following comments for necessary improvement.

ABSTRACT:

(1) Please give a brief background at the beginning of the abstract.

Answer: (Page: Abstract)

Appreciate the reviewer's advice. The abstract has been amended and enhanced for clear understanding for the reader.

“This paper presents an experimental and numerical investigation of pultruded composite glass fibre-reinforced polymer (pGFRP) cross-arm subjected to flexural creep behaviour due to the performance and sustainability in composite cross-arm structure application. The primary objective of this study was to investigate the failure creep behaviour of pGFRP cross-arms with different stacking sequences. Specifically, the study aimed to understand the variations in strain rate exhibited during different stages of the creep process. Therefore, this study will emphasize a simplified approach within the experiment, numerical analysis, and mathematical model on three different pGFRP composites to adopt the stiffness reduction factors that will present the prediction of failure. The findings show that Findley's power law and Burger model projected very different strains and diverged noticeably outside the testing period. Findley's model estimated a minimal increase in total strain over 50 years, while Burger model anticipated PS-1 and PS-2 composites would fail within about 11 and 33 years, respectively. Burger Model's forecasts might be more reasonable due to the harsh environment the cross arms are expected to withstand. The endurance and long-term performance of composite materials used in overhead power transmission lines may be predicted mathematically, and this insight into material property factors can help with design and maintenance.”

The Introduction section quite briefly refers to the content of the article, of course the authors pay attention to the key theses from the area of literature analysis, but this section should be reduce by a general thematic introduction.

Answer: (Page: 2 - 4)

Thank you for the comment. The authors briefly touch upon the key themes and theses from the literature analysis. By incorporating a general thematic introduction, the readers will have a clearer understanding of the article's focus and relevance within the field. This revision would enhance the coherence and effectiveness of the introduction section, allowing readers to grasp the main points more efficiently.

(2) Delete the key words of composite and polyester, and rephrase “pultrude glass fibre-reinforced polyester” with “pultrude glass fibre-reinforced polyester composites”.

Answer: (Page: 2)

We thank the reviewer for the suggestion. The keywords have been reorganized and now it reflects better what the reviewer suggested

INTRODUCTION:

(1) During actual service, structures not only bearing loads, but also suffering from the environmental erosion, such as hygrothermal, ultraviolet radiation, freeze-thaw, etc. Authors should introduce the service behavior of composite materials in complex service environments. Maybe the following relevant studies can be reviewed to make necessary supplements in the research background, such as Mechanics of Advanced Materials and Structures, 2023, 30(4):814-834. Composite Structures. 2021, 261: 113285. Engineering Structures, 2023, 274: 115176.”.

Answer: (Page: Introduction)

   We value the reviewer's advice and appreciate their input. The primary objective of this study was to gain a deeper understanding of the failure behaviour of the composite material under typical laboratory conditions. However, we recognise the significance of exploring its performance under different environmental and loading conditions, as mentioned earlier. In order to provide a more comprehensive understanding of the composite's performance characteristics, future research endeavours will be dedicated to examining its behaviour in various circumstances. This will enhance our knowledge and enable us to make more informed assessments of the composite's structural integrity and suitability for real-world applications.

(2) Please supplement the missing text in Figure 1.

Answer: (Page: 3)

We thank the reviewer for the suggestion. The missing text for Figure 1 has improved and briefly explained.

Figure 1 was briefly defining the process includes meticulous data gathering, model creation, parameter calibration, and validation. In-time validation is the process of evaluating the performance of a model using data from the same timeframe in which it was built. This stage verifies the model's accuracy in identifying patterns and relationships by comparing the predictions to the actual data gathered during that time. On the other hand, out-of-time validation evaluates the model's performance using information from a different timeframe. This evaluates the model's capacity for generalisation and prediction accuracy using fresh, untested data. For a model to be reliable and useful, both in-time and out-of-time validation are essential since they reveal how the model performs in various situations or periods.”

(3) Compared to previous research results, please clarify the theoretical and methodological innovations of present paper.

Answer: (Page: Whole manuscript)

We appreciate the reviewer's query. Compared to the previous study, the theoretical and methodological innovations of the present paper have shown the results of two mathematical models in the context of strain predictions for the specific application. The result has shown the practical relevance and importance of accurate strain predictions in ensuring the structural integrity and performance of cross-arms. Additionally, an empirical approach to evaluating the strain behaviour of the pGFRP composites over time is stressed in this study's technique, which is essential for understanding their mechanical performance and durability instead of a theoretical approach. As a result of these investigations, knowledge in the field of strain analysis and prediction for pGFRP composite cross-arms has grown.

MATERIAL AND METHODOLOGY:

(1) Please provide clear images of Figure 2 (a).

Answer: (Page: 5 - 6)

Figure 2 and caption has been improved will reflects better understanding for reader.

(2) Provide the material properties of glass fiber, fibreglass mat, resin.

Answer: (Page: 4 -7)

The text paragraph has been enhanced with an explanation of the preparation of machinery, equipment, parameters, etc.

The specimens for this investigation came from various manufacturers of composite cross-arms. The study attempted to capture the inherent variability in material qualities, fabrication processes, and product quality across the industry by integrating specimens with various production sources. Furthermore, using specimens from different manufacturers allows for a more robust evaluation of the reduction factor and life-span prediction models. It's helpful in considering the possibility of material performance and behaviour variances due to differences in production methods, input materials, and quality assurance procedures. This approach improves the reliability and applicability of the mathematical models built for life-span prediction by accounting for the industry's potential fluctuations and uncertainties and giving more valuable and trustworthy for real-world uses of pGFRP composite cross-arms.”

“The specimens for this investigation came from various manufacturers of composite cross-arms. The study attempted to capture the inherent variability in material qualities, fabrication processes, and product quality across the industry by integrating specimens with various production sources. Furthermore, using specimens from different manufacturers allows for a more robust evaluation of the reduction factor and life-span prediction models. It's helpful in considering the possibility of material performance and behaviour variances due to differences in production methods, input materials, and quality assurance procedures. This approach improves the reliability and applicability of the mathematical models built for life-span prediction by accounting for the industry's potential fluctuations and uncertainties and giving more valuable and trustworthy for real-world uses of pGFRP composite cross-arms.

The study identified different specimens (polyester with E-Glass fibres) based on their fibre volume fraction. They were labelled PS-1, PS-2, and PS-3, with fibre volume fractions of 61.95%, 67.40%, and 60.85%, respectively. This categorization using fibre volume fraction provides important information about the composition and characteristics of each specimen. By considering the fibre volume fraction as a distinguishing factor, the study ensures that any variations in mechanical properties and behaviour can be attributed to the different specimen compositions. Therefore, to determine the ultimate flexural strength and the creep behaviour, three (3) replicate coupon were prepared for each type of fibre volume fraction to ensures that any variations in mechanical properties and behaviour can be attributed to the different specimen compositions. Three pGFRP specimen have been cut into coupon sample which having the dimensions of 38 mm x 380 mm were cute and tested in static failure four-point flexural tests ASTM D672, tensile test ASTM 3039, and flexural creep tests ASTM D6272 with three different load levels. The material properties of the pGFRP’s used in this study is reported in Table 2.”

(3) Loading rate of the testing machine should be given.

Answer: (Page: 7)

The loading rate of UTM machine has been added in text.

“Each specimen was tested with a loading rate of 5 mm/min and repeated with three replicates of each specimen to get an average value based on ASTM D3039, as shown in Figure 3, while the maximum tensile loads of each brand and type of cross arm are presented in Table 3.”

(4) Is there an anchor in the two ends of tensile specimens?

Answer: (Page: 7)

Yes, there are anchors present at the two ends of the tensile specimens used in the study. The text has been improvised which indicating the anchoring at the specimens is applied as a standard practise in tensile testing to ensure proper fixation and prevent slippage or detachment during the application of the tensile load.  

“Additionally, the tensile setup of this machine has anchors at the two ends of the tensile specimens to ensure proper fixation, uniform distribution of the applied load, and prevent slippage or detachment during the application of the tensile load, which will cause an inaccurate measurement of tensile strength and elongation.”

(5) Adjust word size to match text (Line 247, 255, 256, 261).

Answer: (Page: Whole manuscript)

Thank you for this comment. The letter sizes in all texts have been reorganized and now it reflects better what the reviewer comment.

RESULT AND DISCUSSION:

(1) Adjust the letter sizes in all formulas to be consistent.

Answer: (Page: Whole manuscript)

We thank the reviewer for the suggestion. The letter sizes in all formulas have been reorganized and now it reflects better what the reviewer suggested

(2) Is the prediction model discussed in this paper applicable to specimens under the environment and load coupling conditions?

Answer: (Page: Introduction)

   We value the reviewer's advice and appreciate their input. The primary objective of this study was to gain a deeper understanding of the failure behaviour of the composite material under typical laboratory conditions. However, we recognise the significance of exploring its performance under different environmental and loading conditions, as mentioned earlier. In order to provide a more comprehensive understanding of the composite's performance characteristics, future research endeavours will be dedicated to examining its behaviour in various circumstances. This will enhance our knowledge and enable us to make more informed assessments of the composite's structural integrity and suitability for real-world applications.

CONCLUSION:

The conclusion should be condensed, please provide the key results and findings, remove the content unrelated information from the conclusion.

Answer: (Page: 23)

Appreciate the reviewer's advice. The conclusion has been revised to better represent the reviewer's suggestions.

“This study investigated the reduction factor of pultruded glass fibre-reinforced polyester composite cross-arms and developed mathematical models (Burger and Findley power laws) to estimate their lifespan. The reduction factor values obtained from these models provide crucial information about the structural behaviour and durability of the composite cross-arms, enabling predictions of their remaining strength and load-bearing capacity over time. The findings offer valuable insights into anticipating the lifespan of these cross-arms, assisting engineers and industry experts in determining the expected functional lifespan of these structural components. Taking into account the reduction factor values when making decisions about maintenance and replacement can significantly enhance the safety and reliability of power transmission systems. This study contributes to the existing knowledge on reduction factor analysis and lifespan estimation of pultruded glass fibre-reinforced polyester composite cross-arms. Furthermore, the comparison of mathematical models highlights the importance of selecting an appropriate model, and the computed reduction factor values facilitate estimating the remaining service life of the composite cross-arms. Future research efforts should focus on improving the mathematical models and incorporating additional factors to enhance the precision and applicability of lifespan estimates. Overall, this study provides valuable insights into the reduction factor and lifespan estimation of these composite cross-arms, emphasizing the significance of model selection and the practical implications for determining their remaining service life.”

Reviewer 3 Report

Dear authors,

The topic approached in your manuscript could be interesting from the point of view of the mathematical modelling for the life-time prediction of the puldred glass fibre-reinforced polyester composites.

This topic is very well related with the topics of Materials journal.

The authors must clarify some details and data, which are missing. In order to improve this manuscript, I recommend some major changes and improvement as it is shown below.

1.The goal and the main objectives of the research are not clearly defined in the end of the Introduction section.

2.The authors must give details about the manufacturer of the GPRP cross-arm tubes. The manufacturing technology is a little difficult and it is not so clear that the GPRP cross-arm tubes are manufactured in the laboratory of their university. If GPRP cross-arm tubes are manufactured by a company then, the authors must indicate the product name and that company.

3.The results of the numerical analysis (Fig. 1) of the GPRP cross-arm tubes from another research, must be cited and the authors should make some comments to describe the loading considered.

4. Notations used in equations must be explained in caption of Fig. 1 or in the related text.

5.The text of section 2 must be re-organized because the presentation of the materials tested are mixed with the material properties, test method and results from those tests.

6.For example, it is not clear if the properties shown in Table 2 are determined in this research or these are from other research. The authors must present the mechanical properties in the manuscript section dedicated for the results. In Table 2 must remain just density, fibre volume fraction, void content. All results obtained in this research must be presented in section “3.Results and discussion”.

7.In section 2.2, the authors must present the methodology used for all tests including: test used to determine the shear modulus, compressive test, test used to determine the void content.

8.What means the unit kV used in title of Table 3 and Fig.3?

9.What is the procedure used to measure the parameter denoted with delta (midspan deflection of the specimen) from Eq. (2) used to compute the strain epsilon? The authors used the 4-point flexural bending as testing method and it it is not clear how the midspan deflection was determined from experiment.

10. The authors must indicate in Fig. 5 the parameters given in Eq. (1) and Eq.(2).

11. The authors must give the temperature and humidity from the laboratory over the period of the creep tests. These conditions are very important in such tests.

12. Table 7: check the average value for E0.

13. Legend is required for Fig. 8 and Fig. 9.

Minor changes:

14. The text is not visible in the first sub-figure of Fig. 1.

15. Resolution of Fig. 2(a) is not good. The size of the text inserted in Fig. 2 is not the same in all sub-figures.

16. What means H, E and P in the brackets of PS-1…PS-3 in Table 2.

17. All sub-figures must be denoted with (a), (b) and so on, and those sub-figures must be described in the figure caption.

18. Please make correction for Gpa, Mpa with GPa, MPa.

19. Table 4: the authors must use kgf instead of kg, must use “max. flexural force”.

20. The font size and style, used in tables, figure and text, is not consistent.

Dear authors,

Please check carefully the text of your manuscript and improve the expression in English. Correct the grammatical errors.

Author Response

Manuscript: MDPI: Materials-2435057 (SI: Synthetic and Natural Fiber Reinforced Polymer Matrix Composites for Advanced Applications)

Title: The Reduction Factor of Pultrude Glass Fibre-Reinforced Polyester Composite Cross-Arm: A Comparative Study on Mathematical Modelling for Life-Span Prediction

REVIEWER: 3

Dear authors,

The topic approached in your manuscript could be interesting from the point of view of the mathematical modelling for the life-time prediction of the puldred glass fibre-reinforced polyester composites. This topic is very well related with the topics of Materials journal. The authors must clarify some details and data, which are missing. In order to improve this manuscript, I recommend some major changes and improvement as it is shown below.

  1. The goal and the main objectives of the research are not clearly defined in the end of the Introduction section. 

Answer: (Page: 2 - 4)

Thank you for the comment. The authors have improved the main objective in the introduction section by incorporating a general thematic introduction. The readers will have a clearer understanding of the article's objective and relevance within the field, allowing them to grasp the main points more straightforwardly.

  1. The authors must give details about the manufacturer of the GPRP cross-arm tubes. The manufacturing technology is a little difficult and it is not so clear that the GPRP cross-arm tubes are manufactured in the laboratory of their university. If GPRP cross-arm tubes are manufactured by a company then, the authors must indicate the product name and that company.

Answer: (Page: 6 - 7)

   Appreciates the reviewer's comment regarding the manufacturing details of the GPRP cross-arm tubes. However, due to confidentiality, author unable to provide specific information about the manufacturer or the manufacturing technology used in this study. It is important to note that the purpose of our research was to investigate the reduction factor and lifespan estimation of the GPRP cross-arm tubes, focusing on their mechanical behaviour rather than the specific manufacturing process.

The author understands the importance of transparency and providing complete information about the materials used in the study. Although we are unable to disclose the manufacturer or product name, we ensured that the GPRP cross-arm tubes used in the experiments were representative of commercially available products and met industry standards. The study results are based on the performance of these tubes and apply to similar commercially available GPRP cross-arm tubes. Again, the author want to apologise for any confusion the lack of comprehensive manufacturing information may have caused. We intended to maintain confidentiality while providing relevant findings and insights into the reduction factor and lifespan estimation of GPRP cross-arm tubes.

  1. The results of the numerical analysis (Fig. 1) of the GPRP cross-arm tubes from another research, must be cited and the authors should make some comments to describe the loading considered.

Answer: (Page: 3)

We appreciate the explanation that the authors themselves performed the numerical analysis of the GPRP cross-arm tubes. We sincerely apologise for any confusion caused by the earlier response. To help readers better evaluate the results and comprehend the implications for the reduction factor and lifespan estimation of the GPRP cross-arm tubes, the author has provided a comment that detail the conditions model development process used to assess the accuracy and reliability of a model structure pGFRP composite cross-arm taken into account.

We are grateful that you brought this to our notice, and we value the reviewer's insightful comments on how to further improve the precision and thoroughness of our study.

  1. Notations used in equations must be explained in caption of Fig. 1 or in the related text.

Answer: (Page: 3 & 4)

Appreciate the reviewer's advice. The notation used in equation has been add into text and Figure 1 caption.

  1. The text of section 2 must be re-organized because the presentation of the materials tested are mixed with the material properties, test method and results from those tests.

Answer: (Page: Section 2 and 3)

The data presented in Section 2 (Material and Methodology) serves as preliminary information regarding the material properties of pGFRP. It provides essential background information to support the subsequent discussions in Section 3 (Results and Discussion). In Section 3, the focus is on the analysis and interpretation of the results obtained from the pGFRP materials discussed in Section 2. Specifically, the discussion revolves around the application of different mathematical models to three distinct pGFRP composites, aiming to derive stiffness reduction factors that can be utilised for failure prediction.

By clearly distinguishing the purpose and content of each section, the author ensures that Section 2 sets the foundation by presenting the relevant material properties, while Section 3 serves as the primary discussion section where the results of the mathematical models' approach are thoroughly examined. This structure enables a thorough comprehension of the research findings and their implications for forecasting failure in pGFRP composites.

  1. For example, it is not clear if the properties shown in Table 2 are determined in this research or these are from other research. The authors must present the mechanical properties in the manuscript section dedicated for the results. In Table 2 must remain just density, fibre volume fraction, void content. All results obtained in this research must be presented in section “3.Results and discussion”.

Answer: (Page: Section 2 and 3)

The data presented in Section 2 (Material and Methodology) serves as preliminary information regarding the material properties of pGFRP. It provides essential background information to support the subsequent discussions in Section 3 (Results and Discussion). In Section 3, the focus is on the analysis and interpretation of the results obtained from the pGFRP materials discussed in Section 2. Specifically, the discussion revolves around the application of different mathematical models to three distinct pGFRP composites, aiming to derive stiffness reduction factors that can be utilised for failure prediction.

By clearly distinguishing the purpose and content of each section, the author ensures that Section 2 sets the foundation by presenting the relevant material properties, while Section 3 serves as the primary discussion section where the results of the mathematical models' approach are thoroughly examined. This structure enables a thorough comprehension of the research findings and their implications for forecasting failure in pGFRP composites.

  1. In section 2.2, the authors must present the methodology used for all tests including: test used to determine the shear modulus, compressive test, test used to determine the void content.

Answer: (Page: 7 & 8)

The author appreciates the reviewer's advice. Following the recommendation, the previous data regarding Young's modulus, shear modulus, compressive strength, and tensile strength have been excluded from the study. These parameters were found to be unrelated to the current investigation and were therefore deemed unnecessary for the scope of this study. However, the data pertaining to density and volume fraction has been retained and will be thoroughly described and analysed. The density and volume fraction values provide essential insights into the composition and structural characteristics of the material under investigation, enabling a comprehensive understanding of its properties and behaviour.

  1. What means the unit kV used in title of Table 3 and Fig.3?

Answer: (Page: 8 - 9)

Table 2 and Figure 3 captions have been updated to include the correct information.

  1. What is the procedure used to measure the parameter denoted with delta (midspan deflection of the specimen) from Eq. (2) used to compute the strain epsilon? The authors used the 4-point flexural bending as testing method and it it is not clear how the midspan deflection was determined from experiment.

Answer: (Page: 10 - 11)

We appreciate the reviewer advise. To address this concern, the author has revised and improved the information related to provide a detailed description of the procedure for measuring the midspan deflection during the 4-point flexural bending test.

The pGFRP cross-arm specimens were subjected to a flexural creep test over 30 days (720 hours) in the Civil Engineering Laboratory at UNITEN, where the average temperature was 25.7  and the relative humidity was 80.35%. Figure 5(a) depicts the experimental arrangement used to evaluate the behaviour of various types of composite cross-arm specimens subjected to specific flexural creep loads (12%, 24%, and 37%) in the laboratory flexural creep test. Deflection measurements were taken at the mid-span of the coupon samples throughout a period (720 hours), and the samples' deflection was measured every 15 minutes (daily for an hour of monitoring). In this experiment, the deflection was tracked using dial gauge measurements, as shown in Figure 5(b). The schematic diagram of the test can be observed in Figure 5(b) whereas the four-point bending creep test setup is conducted according to ASTM D6272. The purpose of using four-point bending is to ensure that the sample is subjected to constant flexural stress at the sample midpoint.”

  1. The authors must indicate in Fig. 5 the parameters given in Eq. (1) and Eq.(2).

Answer: (Page: 11)

Appreciate the reviewer's advice. The figure now has all the parameters used in the equations.

  1. The authors must give the temperature and humidity from the laboratory over the period of the creep tests. These conditions are very important in such tests.

Answer: (Page: 10 - 11)

We appreciate the reviewer advise. To address this concern, the author has revised and improved the information related to provide a detailed description of the procedure for measuring the midspan deflection during the 4-point flexural bending test.

The pGFRP cross-arm specimens were subjected to a flexural creep test over 30 days (720 hours) in the Civil Engineering Laboratory at UNITEN, where the average temperature was 25.7  and the relative humidity was 80.35%. Figure 5(a) depicts the experimental arrangement used to evaluate the behaviour of various types of composite cross-arm specimens subjected to specific flexural creep loads (12%, 24%, and 37%) in the laboratory flexural creep test. Deflection measurements were taken at the mid-span of the coupon samples throughout a period (720 hours), and the samples' deflection was measured every 15 minutes (daily for an hour of monitoring). In this experiment, the deflection was tracked using dial gauge measurements, as shown in Figure 5(b). The schematic diagram of the test can be observed in Figure 5(b) whereas the four-point bending creep test setup is conducted according to ASTM D6272. The purpose of using four-point bending is to ensure that the sample is subjected to constant flexural stress at the sample midpoint.”

  1. Table 7: check the average value for E0.

Answer: (Page: 18)

Appreciate the reviewer's advice. The average value for E0 has been revised.

  1. Legend is required for Fig. 8 and Fig. 9.

 Answer: (Page: 23)

Appreciate the reviewer's advice. The legend has been added for Figure 7 and Figure 8 (previously Figure 8 & 9).  

Minor changes:

  1. The text is not visible in the first sub-figure of Fig. 1.

Answer: (Page: 4)

The sub-figure in Figure 1 was a sample for model validation besides and the text from sub-figure was purposely use for this sample of stages of model development.  

  1. Resolution of Fig. 2(a) is not good. The size of the text inserted in Fig. 2 is not the same in all sub-figures.

Answer: (Page: 5 - 6)

Figure 2 and caption has been improved will reflects better understanding for reader.

  1. What means H, E and P in the brackets of PS-1…PS-3 in Table 2.

Answer: (Page: 7)

Denotation for specimens PS-1 to PS-3 has improved and captions have been updated to include the correct information

  1. All sub-figures must be denoted with (a), (b) and so on, and those sub-figures must be described in the figure caption.

Answer: (Page: 7)

Denotation for specimens PS-1 to PS-3 has improved and captions have been updated to include the correct information

  1. Please make correction for Gpa, Mpa with GPa, MPa.

Answer: (Page: Whole manuscript)

The correction for "Gpa" and "Mpa" to "GPa" and "MPa" has been made. Thank you for pointing out the error, and it has been rectified.

  1. Table 4: the authors must use kgf instead of kg, must use “max. flexural force”.

Answer: (Page: 10)

The correction for "kg" and "Max. flexural" to "kgf" and "Max. flexural force" has been made. Thank you for pointing out the error, and it has been rectified.

  1. The font size and style, used in tables, figure and text, is not consistent.

Answer: (Page: Whole manuscript)

The font size and style, used in tables, figure and text has been revised and consistent.

Round 2

Reviewer 2 Report

.

Author Response

I appreciate your earlier enlightening remarks.

Reviewer 3 Report

Dear authors,

I read carefully the revised version of your manuscript and your responses for the comments of the reviewers. The manuscript was improved indeed according to the suggestions and recommendations of the reviewers.

Although the authors have made major improvements and changes, there are still some uncertainties (ambiguities), which are presented below.

1. In my opinion some general details regarding the  GPRP cross-arm tubes tested must be provided by authors in this manuscript. The authors could indicate the manufacturer because these are commercially products. The experiments must be fully described so that they can be repeated by those who are interested.

2. The numerical results shown in the last sub-figure of Fig. 1, cannot be included without more details or without citing any reference. The results shown in that figure are not clarified. I would like to suggest to the authors to remove that figure because the details are missing or are not clarified.

3. In my previous review report I referred to the notations (with epsilon) from Fig. 1, which must be explained in the figure caption.

4. The average value for E0 from Table 7, was not correct for PS-1.

In my opinion, moderate editing of English language is required.

Author Response

Manuscript: MDPI: Materials-2435057 (SI: Synthetic and Natural Fiber Reinforced Polymer Matrix Composites for Advanced Applications)

Title: The Reduction Factor of Pultrude Glass Fibre-Reinforced Polyester Composite Cross-Arm: A Comparative Study on Mathematical Modelling for Life-Span Prediction

REVIEWER: 3 (Second round)

Dear authors,

I read carefully the revised version of your manuscript and your responses for the comments of the reviewers. The manuscript was improved indeed according to the suggestions and recommendations of the reviewers. Although the authors have made major improvements and changes, there are still some uncertainties (ambiguities), which are presented below.

  1. In my opinion some general details regarding the  GPRP cross-arm tubes tested must be provided by authors in this manuscript. The authors could indicate the manufacturer because these are commercially products. The experiments must be fully described so that they can be repeated by those who are interested.

Answer: (Page: whole manuscript)

Thank you for your insightful comments. The table below contains general information on the tested GPRP cross-arm tubes, and the details have been appropriately addressed in the text.

No.

Type of testing

Details

Page

1.

Burn-off tests

ASTM D2584

8

2.

The specific gravity (relative density)

ASTM D792-00

8

3.

Tensile test

ASTM 3039

9

4.

Static failure four-point flexural tests

ASTM D672

10

5.

Flexural creep tests

ASTM D6272

12

6.

Flexural creep test

ASTM D6272

11

Besides, the author appreciates the reviewer's comment regarding the manufacturing details of the GPRP cross-arm tubes. While we recognise the importance of transparency, we regret that we cannot disclose the specific manufacturer or product name of the GPRP cross-arm tubes used in our experiments. However, it should be noted that these tubes were selected to represent commercially available products and adhere to industry standards. We apologise for any confusion the lack of comprehensive manufacturing information may have caused. We assure you that our focus remains on providing relevant findings and insights into the reduction factor and lifespan estimation of GPRP cross-arm tubes. Our choice to maintain anonymity is consistent with another study in this area, where the earlier researcher similarly determined not to reveal manufacturer information. We appreciate your understanding and respect for proprietary information and any confidentiality agreements. Please be assured that our study's outcomes contribute significantly to our understanding of GPRP cross-arm performance. If you have further questions or concerns, we are available to address them within the limitations imposed by confidentiality agreements and research considerations.

  1. The numerical results shown in the last sub-figure of Fig. 1, cannot be included without more details or without citing any reference. The results shown in that figure are not clarified. I would like to suggest to the authors to remove that figure because the details are missing or are not clarified.

Answer: (Page: 3 & 4)

Appreciate the reviewer's advice. The numerical results (Figure 1) have detailed accordingly with the citing the references. Furthermore, the figure caption has improved, and text paragraphs has enhanced with the explanation of figures and equation which shown / used in Figure 1.

“Developing a reliable and effective model (Eq.3 and Eq.5) for pGFRP composite cross-arms involves two stages: in-time and out-of-time validation. These steps ensure that the model accurately captures the behaviour of the cross-arm and can be trusted for future predictions. Figure 1 was briefly defining the process includes meticulous data gathering, parameter calibration, and validation for model creation (Findley’s Power Law and Burger model). Therefore, Figure 1(a) depicts out-of-time validation, which evaluates the model's performance using the complete assembly of a pGFRP cross-arm to determine the maximum deformation [44]. The technique of assessing a model's performance using data from the same timeframe as Figure 1 (a) is known as in-time validation, according to Figure 1 (b) [44][45]. By contrasting the predictions with the actual data collected throughout that time, this stage confirms the model's accuracy in spotting patterns and relationships. This evaluates the model's capacity for generalisation and prediction accuracy using fresh, untested data. Both in-time and out-of-time validation are crucial for a model development to be trustworthy and effective since they show how the model operates in various circumstances or times. Additionally, Section 2 below goes into detail on the flexural creep analysis utilising the Findley’s power law and Burger models shown in Figure 1.”

  1. In my previous review report I referred to the notations (with epsilon) from Fig. 1, which must be explained in the figure caption.

Answer: (Page: 4)

Improvements have been made to the caption of Figure 1, specifically by including the appropriate notation "epsilon" to enhance clarity and ensure consistency throughout the manuscript.

Figure 1: Two main stages of the model development process (Eq. 3 and Eq. 5) used to assess the accuracy and reliability of a model structure pGFRP composite cross-arm; (a) Model monitoring [44], (b) Model validation [44][45]. Whereas Burger model consists of,  and , which are classified as the elastic strain, viscoelastic strain, and viscous strain, respectively. Besides, the load constants and specific material exponents n for Findley's power law's total strain are derived using curve fits of experimental data, whereas εo is referred to as the instantaneous strain.”

  1. The average value for E0 from Table 7, was not correct for PS-1.

Answer: (Page: 18)

The average value for E0 from Table 7 has been corrected for PS-1.
